# The Importance of the Interventional Pathologist in Fine Needle Aspiration and Core Needle Biopsy Procedures for Lymph Node Lesions: A Retrospective Analysis of Diagnostic Correlation and Proposal of a Classification System for Lymph Node Core Needle Biopsies

**DOI:** 10.3390/diagnostics15202628

**Published:** 2025-10-17

**Authors:** Montserrat de la Torre Serrano, Ana María Colino-Gallardo, Jesús Vega González, Maria Reyes Bergillos Giménez, Ramón Robledano Soldevilla, Teresa Iscar Galán, Julian Sanz Ortega, Maria del Mar Olmo Fernández, Santiago Nieto Llanos, Karen Villar Zarra

**Affiliations:** 1Department of Pathology, Clínica Univesidad de Navarra, 28027 Madrid, Spaintiscarg@unav.es (T.I.G.); jsanzo@unav.es (J.S.O.); 2Department of Pathology, Hospital Universitario Infanta Sofía, San Sebastián de los Reyes, 28702 Madrid, Spain; 3Department of Pathology, Hospital Universitario Clínico San Carlos, 28040 Madrid, Spain; jesusvega.gon@gmail.com; 4Department of Pathology, Hospital Universitario del Henares, Coslada, 28822 Madrid, Spainkarenvillarz@gmail.com (K.V.Z.)

**Keywords:** cytopathology, interventional pathologist, fine needle aspiration biopsy (FNA), core needle biopsy (CNB), ultrasound, minimally invasive

## Abstract

**Introduction**: Pathology requires the integration of macroscopic and microscopic findings for an accurate diagnosis. Fine needle aspiration biopsy (FNA) traditionally became the domain of radiologists with the introduction of ultrasound; however, its increased accessibility has allowed other specialists, including pathologists, to incorporate it into their daily practice. Both ultrasound-guided FNA and core needle biopsy (CNB) performed by interventional pathologists have proven valuable tools in the diagnosis of lymph node lesions. **Materials and Methods**: An observational, descriptive, retrospective study was conducted at Hospital Universitario del Henares, analysing 134 FNABs and 31 CNBs of lymph node lesions between 2023 and 2024. The diagnostic yield of both techniques and their correlation were evaluated. **Results**: Ultrasound facilitated better lesion characterization. FNA demonstrated a diagnostic yield of 98%, with a 94% correlation with CNB results. The Sydney system 2020 was applied to FNA reporting in lymphatic node. An equivalent classification system for CNB is proposed. **Conclusion**: The performance of FNA and CNB by pathologists optimizes diagnosis, reduces time and costs, and strengthens their role in minimally invasive diagnostics. Adoption of a unified classification system for CNB lesions would facilitate communication between specialists.

## 1. Introduction

To understand disease, our ancestors began studying organs from cadavers to learn about the causes and development of pathological processes. Careful macroscopic examination of these organs was essential for gaining such knowledge, so accurate descriptions were crucial. Notable figures like Leonardo da Vinci made important contributions to this work [1].

Pathological Anatomy began to establish itself as a scientific discipline in the latter half of the 19th century, with the incorporation of the microscope [1]. Our specialty thus evolved to combine macroscopic diagnosis with histological examination, initiating the study of disease at the cellular and tissue level, a theory introduced by Rudolf Virchow in 1858 [1,2]. With the subsequent advancement of surgical techniques, histological studies expanded to include not only cadaveric tissue but also surgical specimens, which, over time and with the progression of medical science, have evolved into the analysis of increasingly smaller biopsy samples [3].

A further significant development in our field emerged initially in the United Kingdom and the United States, where a novel diagnostic method was devised and later widely adopted and standardised during the 1950s and 1960s by our Swedish and Dutch colleagues [3]. This technique involved the aspiration of tumours and solid organs using a fine needle, for subsequent cytological evaluation, a method now known as fine needle aspiration (FNA) [3]. As a result, the term “interventional pathologist” or “interventional cytopathologist” was coined [4,5,6].

Initially, fine-needle aspirations were limited to superficial and palpable lesions. However, with advancements in medicine and diagnostic imaging techniques, ultrasound emerged as a means of accessing non-visible lesions [4]. As a consequence, one of our competencies began to fall under the remit of those employing this technology, namely, radiologists [4,6].

Nevertheless, as medical technology advanced and the cost of ultrasound equipment decreased, this tool became increasingly accessible to other medical specialists in their routine clinical practice, including the performance of interventional procedures with an acceptable level of patient safety [3,6,7].

The increasing availability of portable ultrasound equipment, combined with the global trend toward decentralized diagnostics, has further emphasized the need for specialists who can perform and interpret these procedures at the point of care.

Consequently, today, even non-medical healthcare professionals, such as physiotherapists, utilize this technology to enhance patient care and management; ultimately, the patient remains our primary focus.

Because of this evolution, the term POCUS (Point-of-Care Ultrasound) was introduced, referring to ultrasound performed in real-time by a healthcare professional [5,6,7]. As healthcare providers and specialists in the macroscopic diagnosis of lesions since the origins of our discipline, the integration of ultrasound into our diagnostic process is both logical and justified. The ultrasound image provides macroscopic information about the lesion, an essential component for accurate diagnostic correlation [7,8,9,10].

Moreover, our specialty is both generalist and integrative in nature, positioning us in an optimal scenario to transform our field by adapting to emerging technologies. This is already evident in the implementation of digital pathology and molecular diagnostics, both of which contribute towards achieving more precise and personalised medicine [5,7].

In this context, the pathologist, traditionally perceived as a “laboratory” professional, has progressively assumed a more visible and proactive clinical role. For FNA guided by ultrasound to be feasible in a routine interventional pathology consultation, the lesions must be superficially located to ensure its access without the need for operating rooms or special procedural suites. This requirement ensures that the procedure can be performed efficiently and safely in an outpatient environment, facilitating timely diagnosis and patient management.

By performing procedures such as FNA, the pathologist becomes an integral part of multidisciplinary teams, providing immediate input into diagnostic and therapeutic decisions. This shift not only enhances the efficiency of diagnostic pathways, but also fosters closer communication with clinicians and contributes to a more holistic approach to patient care. FNA is an increasingly established diagnostic procedure within hospitals, offering immediate diagnosis in a rapid, reproducible, and minimally invasive manner. This translates into greater patient comfort and does not compromise the possibility of subsequent surgical intervention [3,7,8,11]. Numerous studies support the higher diagnostic yield of ultrasound-guided FNA when performed by experienced pathologists compared to other specialists [3,4,7,10,11,12]

On one hand, the efficiency of the procedure is enhanced by involving a single specialist rather than at least two, as the pathologist is capable of performing the sampling, triaging the material, and interpreting it in real time [6]. Furthermore, the effectiveness of the process is increased, given that the pathologist is uniquely equipped to integrate the macroscopic features of the lesion with the expected microscopic findings [6,8,10]. For example, a hypoechoic, well-defined, rounded lesion with posterior acoustic enhancement may suggest a cyst on ultrasound. However, a homogeneous, solid lesion under microscopy, such as a metastatic melanoma, the “great mimicker”, may present with an identical ultrasound appearance.

In addition, the pathologist is the only professional formally trained to perform and interpret ROSE (Rapid On-Site Evaluation), which directly impacts subsequent clinical decision-making [5,6,7,9,10,11,13]. It is worth emphasising that a diagnosis of cancer cannot be confirmed without a histopathological report [5].

These advantages are equally transferable to patients, who benefit from fewer hospital visits and an expedited diagnosis, avoiding prolonged waiting times, as well as to hospitals, which can optimise appointment scheduling and resource management. Other specialists, recognising these benefits, are increasingly referring patients to our services [5,7,10,11].

Nonetheless, FNA is not without diagnostic limitations [4]. In certain cases, diagnosis requires the acquisition of additional material to enable the application of ancillary techniques such as immunohistochemistry or molecular testing, both of which necessitate a minimum number of tumour cells. Furthermore, some diagnoses rely on architectural features, such as evidence of tissue invasion, that can only be adequately assessed with a biopsy specimen. In many instances, the cell block obtained from FNA is insufficient to resolve such diagnostic challenges.

This need prompted a further advancement in the specialty through the incorporation of core needle biopsy (CNB). Ultrasound-guided CNB is technically similar to FNA, and therefore, a well-trained pathologist can readily acquire this skill as a complementary aspect of routine practice. This further enriches the discipline and positions pathologists as leaders in minimally invasive diagnostics [4,7].

These developments also have educational and institutional implications. The inclusion of ultrasound-guided techniques in pathology residency programs, already implemented in several countries, strengthens the autonomy and diagnostic capacity of future specialists. Moreover, the clinical relevance of FNA and CNB in oncology, infectious diseases, and inflammatory conditions reinforces the indispensable role of the pathologist in modern multidisciplinary teams. In a healthcare system under increasing pressure to optimise resources and reduce diagnostic delays, our ability to perform minimally invasive procedures with high diagnostic accuracy contributes meaningfully to both patient outcomes and institutional efficiency. This makes a strong case for the formal recognition and expansion of interventional pathology as a core competency within Anatomical Pathology.

The present study aims to evaluate the diagnostic yield of FNA in superficial lymph nodes, one of the most frequent indications in interventional pathology consultations, as well as to assess CNB and the diagnostic correlation between the two techniques.

FNA results from lymph nodes were analysed according to the 2020 Sydney System (LN-FNA) [14], and a similar classification system is proposed for lymph node CNB.

## 2. Materials and Methods

This original research article corresponds to an observational, descriptive, cross-sectional, and retrospective study based on the results of fine needle aspirations (FNA) and core needle biopsies (CNB) of superficial lymph nodes, performed during routine clinical practice in the Department of Pathology at Hospital Universitario del Henares, Coslada, Madrid, Spain. This hospital provides healthcare services to approximately 170,000 inhabitants across five municipalities in the Community of Madrid. The interventional pathology consultation is led by a single pathologist who performs both FNA and CNB, always assisted by a technician. The technical assistance is provided on a rotating basis among those with appropriate training. The consultation is located within the same department, just a few meters from the pathologist’s office, which contributes to more efficient time management and facilitates combining it with microscopic diagnostic work.

Fine needle aspiration (FNA) was performed primarily using Microlance 23G (0.6 × 25 mm) and 22G (0.7 × 30 mm) needles (Becton Dickinson, Franklin Lakes, NJ, USA), and HS CHIBA EcoJEKT needles (23G × 150 mm; Hakko Co., Tokyo, Japan) for deeper nodules. Core needle biopsies (CNB) were obtained using MISSION and MARQUEE disposable core biopsy instruments (Becton Dickinson, Franklin Lakes, NJ, USA), ranging in gauge size from 12G to 18G and in length from 10 to 13 cm. The choice of needle gauge was based on the anatomical location, lesion characteristics, and diagnostic objectives. MISSION devices were used in combination with a coaxial introducer system. Local anesthesia was administered along the biopsy trajectory using Mepivacaine (20 mg/mL) and Epinephrine (0.01 mg/mL) (1.8 mL vials; Normon Laboratories, Madrid, Spain) delivered with a BD Eclipse 25G × 5/8” needle and BD Microlance 23G needle (Becton Dickinson, Franklin Lakes, NJ, USA). The skin incision site (approximately 2–3 mm) was closed using STERI-STRIP 12 mm blue adhesive strips (3M, St. Paul, MN, USA) and covered with a self-adhesive white dressing. Ultrasound scanning was performed using a LOGIQ P9 ultrasound system equipped with an L4–12t-RS linear transducer operating at 10–12 MHz (GE Healthcare, Chicago, IL, USA). Flow cytometry analyses were performed externally (out-house), and therefore, detailed information on the specific equipment used is not available.

A total of 134 FNAs (69 from 2023 and 65 from 2024) and 31 CNBs (16 from 2023 and 15 from 2024) were analysed. The selected periods were chosen to provide the most recent available data on this diagnostic practice carried out by an interventional pathologist at a national reference centre for this procedure. Patients included in the study presented with superficial lymph node lesions in any anatomical region, excluding intrathoracic and intra-abdominal locations, and underwent ultrasound-guided FNA and/or CNB during the aforementioned years. The most frequently analysed anatomical site was the head and neck region. It is worth noting that the majority of CNBs performed on lymph nodes had a previously suspected metastasis based on FNA. In this context, certain immunohistochemical techniques, such as PD-L1, which are therapeutic targets, could be anticipated to begin exploring different treatment options.

For the FNA diagnosis of lymph node lesions in our sample, the Sydney System (LN-FNA), established in 2020, was employed. This system categorizes the findings into five levels: Category I (non-diagnostic), Category II (benign), Category III (atypia of undetermined significance), Category IV (suspicious for malignancy), and Category V (malignant) [14]. In 2024, the World Health Organization (WHO) published the *WHO Reporting System for Lymph Node, Spleen, and Thymus Cytopathology*, which became available in print format in January 2025. This new system also establishes five diagnostic categories that are largely overlapping with those defined by the Sydney System [15]. Given this similarity, and considering that our study includes patient samples from both 2024 and 2023, prior to the publication of the WHO system, all lymph node lesions evaluated by FNA in this study were classified according to the 2020 Sydney System and are presented as such in this study.

In the case of CNB of lymph node, since no specific categorization system had been established, the authors decided to use as categories non-diagnostic, benign, atypical, suspicious for malignancy, and malignant, equivalent to those employed in FNA, in order to improve the communication with the clinicians to whom the reports were addressed.

These procedures were carried out by a trained interventional pathologist with extensive experience in ultrasound-guided sampling, ensuring a consistent and high-quality technique throughout the study period. All aspirations and biopsies were performed using high-frequency linear transducers, and specimen adequacy was assessed in real time using the ROSE method when feasible. This workflow allowed for the immediate triage of material for additional ancillary studies such as immunocytochemistry, flow cytometry, or molecular testing when indicated.

To this end, a review of histopathological reports was conducted using the hospital’s pathology data management system. Clinical information was collected in a dissociated database to which only the research team had access, and all data were pseudonymised.

Statistical analysis was performed using IBM SPSS Statistics for Mac, version 29.0 (IBM Corp., Armonk, NY, USA). A descriptive study of categorical variables was conducted using frequencies and percentages, including: the diagnostic category of FNA according to the 2020 Sydney system, performance of CNB, performance of flow cytometry, and presence of diagnostic concordance between FNA and CNB or between FNA and flow cytometry.

Subsequently, cases in which CNB or flow cytometry had been performed were selected to assess diagnostic concordance. For this purpose, a new dichotomous variable (positive FNA) was created, defined as positive cases classified as suspicious for malignancy (category IV) or malignant (category V) according to the Sydney system, and negative categories from I to III.

Diagnostic concordance was analysed using contingency tables, considering CNB or flow cytometry as the reference tests. From these tables, classical diagnostic performance metrics were calculated: sensitivity, specificity, positive predictive value (PPV), and negative predictive value (NPV). The association between variables was assessed using Pearson’s Chi-squared test, and Fisher’s exact test was applied when expected frequencies were low. A *p*-value < 0.05 was considered statistically significant.

In accordance with Spanish data protection regulations, all clinical data were anonymized. This study was approved by the hospital’s Research Ethics Committee (approval number: 25/257-E, approval date: 27 March 2025).

## 3. Results

Thirteen different clinical departments referred patients for evaluation, with the highest numbers of referrals coming from Internal Medicine (30%), Otorhinolaryngology (23%), and Haematology (15%) (Figure 1). Among the patients, 50% had no prior diagnosis, compared to 35% who did, and 14% were being investigated due to other findings. Patients were questioned regarding the use of antithrombotic medication; 74% reported not taking any, while 25% did, of whom 15% required temporary discontinuation of the medication prior to the procedure (Figure 2). No complications or adverse events were recorded in any patient.

Ultrasound examination was performed in 98% of cases using a linear transducer (L4-12t-RS), with key ultrasound features documented in the reports, including echogenicity, shape, margins, relationship with vascular structures, colour Doppler assessment, and measurement of all three diameters (anteroposterior, lateral, and transverse), with lateral diameter being the most frequently enlarged (46%).

These imaging findings facilitated precise lesion targeting while simultaneously providing crucial information for risk stratification, particularly concerning the evaluation of lymph nodes and soft tissue masses. The application of real-time ultrasound guidance ensured safe access, enhancing overall sample quality and reducing procedural risks. Furthermore, the integration of clinical, radiological, and cytological data proved indispensable in guiding subsequent diagnostic procedures and therapeutic planning.

For FNA, a 23G Microlance blue needle was used in 82% of cases (Figure 3). The mean number of passes was 1 (including passes for flow cytometry), and the average number of slides prepared was 3 (Table 1). Adequate material for diagnosis was obtained in 99% of cases for both FNA and CNB (Figure 4). In the case of CNBs, sufficient material was obtained for immunohistochemical studies in 94% and for molecular studies in 71% (Figure 4). Results were reported using the 2020 Sydney System classification for FNA of lymph-node and in five equivalent categories for CNB (Figure 5).

The distribution of FNA diagnostic categories for lymph nodes was as follows: non-diagnostic (Sydney System Category I) in 2% of cases (*n* = 2), benign (Sydney System Category II) in 47% (*n* = 63), atypia of undetermined significance (Sydney System Category III) in 3% (*n* = 4), suspicious for malignancy (Sydney System Category IV) in 5% (*n* = 7), and malignant (Sydney System Category V) in 43% of cases (*n* = 58) (Figure 5 and Table 1).

CNB was performed in 23% of patients (*n* = 31), while 77% (*n* = 103) did not require CNB following FNA. Flow cytometry was carried out in 59% of cases (*n* = 79), all of them cases with suspected lymphoproliferative process after analysing the FNA at the microscope. It was not performed in 26% (*n* = 35), and was considered not applicable in 15% (*n* = 20).

For CNB procedures, a 16G needle was used in 84% of cases, an 18G in 10%, and a 14G in 6% (Figure 6). The mean number of passes was 3. Rapid on-site evaluation (ROSE) using the touch print technique was performed in 100% of CNBs (Figure 7), and as mentioned before there was sufficient material for complementary studies (Table 2 and Figure 4).

Among patients who underwent CNB (*n* = 31), comparison with the FNA results, dichotomized as positive (Sydney system categories IV-V) or negative (categories I-III), demonstrated a high diagnostic concordance. Among the 30 CNB-positive cases, FNA correctly identified 28 as positive and misclassified 2 as negative. In the 1 CNB-negative case, FNA produced no false positives and correctly classified it as negative (Table 3).

Statistical analysis using Pearson’s Chi-squared test revealed a significant association between both tests (χ^2^ = 19.95; *p* < 0.001), confirming the diagnostic robustness of FNA in this context. In terms of diagnostic performance, FNA achieved a sensitivity of 93.3%, a specificity of 100%, PPV of 100%, and an NPV of 33.3%. These results indicate that a positive FNA result was always confirmed by CNB, whereas negative cases showed diagnostic limitations, reflected in the lower NPV.

Regarding diagnostic concordance, in the cases where flow cytometry was performed (*n* = 79), agreement with FNA results was observed in 91% (*n* = 72), whereas discordance was recorded in 5.2% (*n* = 7). Similarly, among patients who underwent CNB (*n* = 31), concordance between FNA and CNB was achieved in 94% (*n* = 29), while discordance occurred in 1.5% (*n* = 2) (Figure 8).

In these cases, a different pattern was observed. Of the 28 flow cytometry-positive cases, FNA classified 26 as positive and 2 as negative. Among the 51 flow cytometry-negative cases, FNA classified 5 as positive and 46 as negative (Table 4).

Statistical analysis with Pearson’s Chi-squared test did not reach significance (χ^2^ = 3.34; *p* = 0.068), indicating that the association between FNA and flow cytometry was not significant in this series. Diagnostic performance showed a high PPV (83.9%), meaning that when FNA was positive, there was a strong probability of confirmation by flow cytometry. In addition, FNA demonstrated a sensitivity of 92.9%, a specificity of 90.2%, and an NPV of 95.8%, reflecting a good ability of FNA to reliably identify both positive and negative cases when compared with flow cytometry.

Finally, when FNA results were dichotomized into negative (categories I–III) and positive (categories IV–V), 51.5% of the samples were classified as negative (*n* = 69), whereas 48.5% were classified as positive (*n* = 65).

It is worth noting that, although fine needle biopsy (FNB) which uses needle gauges similar to those of FNA, is starting to be implemented in some centres, it is not performed in our setting.

## 4. Discussion

As demonstrated by the results of this study, interventional pathology is an increasingly recognised and requested service among other hospital specialties. By performing the procedures ourselves, we were able to carry out both ROSE during FNA and on-site evaluation with touch-print ROSE during CNB. This enabled rapid assessment of the obtained material, ensuring both its quantity and quality, and guiding subsequent passes to the area of interest. Consequently, we obtained sufficient material with a minimal number of slides, suitable for both diagnosis and ancillary techniques, with excellent diagnostic correlation between FNA and CNB, including flow cytometry in cases of lymphoproliferative processes in lymphatic nodes.

This study shows the diagnostic performance of FNA in lymph nodes, evaluated through its correlation with CNB and flow cytometry. The analysis of 134 FNA procedures classified according to the Sydney system showed a predominance of benign and malignant categories, with nearly half of the cases corresponding to Category II (benign) and a similar proportion to Category V (malignant). These results underscore the clinical relevance of FNA as an initial diagnostic tool in the assessment of lymph nodes, which clinically require evaluation to establish their nature.

The correlation between FNA and CNB demonstrated a high level of diagnostic agreement, with statistical analysis confirming a significant association. Importantly, FNA achieved high sensitivity (93.3%) and perfect specificity (100%), along with a PPV of 100%. These findings highlight the reliability of a positive FNA result, which was invariably confirmed by CNB. On the other hand, the relatively low NPV (33.3%) should be interpreted with caution, as it may be influenced not only by the limitations of FNA in detecting certain malignant entities, but also by the very small number of true negative cases in our series, so this result might not be generalizable.

In contrast, the comparison of FNA with flow cytometry showed different outcomes. Although the association between both tests did not reach statistical significance, FNA demonstrated a high PPV (83.9%), as well as high sensitivity (92.9%) and specificity (90.2%) when flow cytometry was used as the reference standard. Moreover, the NPV (95.8%) suggests that negative cytology results correlated well with negative flow cytometry findings. The absence of statistical significance may be explained by the uneven distribution of the sample (with a predominance of negative results) and the heterogeneity of the lesions studied, which likely reduced the power to detect stronger associations.

Taken together, these findings reinforce the value of FNA as a first-line diagnostic procedure for lymph nodes. Its high PPV supports its role in patient management by enabling rapid confirmation of malignancy and avoiding unnecessary invasive procedures and align with the broader trend in pathology toward minimally invasive diagnostics and personalized medicine.

Although not the primary focus of this study, our center has extensive experience in performing FNA of superficial extra-nodal lesions in the outpatient setting. The technique employed does not differ from those previously described, and the diagnostic yield, achieved with a minimal number of passes and slides, is comparable to the results reported here. These lesions are mainly soft tissue in nature, predominantly benign, such as lipomas and cysts, but also include cutaneous metastases from neoplasms originating in other locations, such as breast cancer. This highlights the broad scope of our practice and suggests that analogous five-tiered classification systems could be developed for FNA superficial extra-nodal lesions.

However, it is important to highlight that, due to the study design, CNBs were only performed in patients with suspicious or malignant cytological results. This introduces a verification bias, preventing the calculation of sensitivity, specificity, and negative predictive value. Despite this limitation, the findings are consistent with previous studies supporting the diagnostic capability of FNA when directly performed by pathologists. Another notable limitation of this study is the limited number of CNB samples available. This may be partly due to clinicians favouring surgical intervention over CNB when there is a strong suspicion of, for example, lymph node metastasis, potentially reflecting a lack of familiarity with the applicability of CNB in such cases. We anticipate that, as awareness and understanding of this technique grow among clinical colleagues, the use of CNB will increase in appropriate cases, thereby offering patients a less invasive and safer diagnostic alternative. Regarding flow cytometry in cases of lymphoproliferative processes, although a good correlation with FNA was observed, the lack of statistical significance may be attributed to the limited sample size and the heterogeneity of the evaluated pathologies. While the majority of our diagnoses were benign, a substantial proportion, nearly equivalent to this category, were either malignant or suspicious. This underscores the value of our interventional pathology consultation, as it helps avoid more invasive procedures in benign lesions while streamlining the diagnostic process for those requiring treatment or surgery. Antithrombotic therapy did not delay scheduling in our clinic, as over 85% of patients did not require its discontinuation, thereby supporting the efficiency of this clinical pathway.

The main advantage of established diagnostic classification systems is that they enable pathologists, even across different countries, to communicate using a standardised language, both among themselves and with other medical specialists. Given the excellent diagnostic concordance observed between FNA and CNB in our study, we propose the development of a five-tiered classification system for lymph node CNBs. Such a framework would enhance consistency, comparability, and clarity in reporting, thereby supporting more accurate clinical decision-making. This proposal aligns with systems already well integrated into daily practice, such as the Paris System 2022 for urinary cytology, the Bethesda System 2023 for thyroid cytopathology, and the Milan System 2020 for salivary gland cytopathology. Although the latter two employ six categories, they essentially follow a closely overlapping model. While originally designed for cytology, these systems can be viewed more broadly as classification schemes for minimally invasive procedures. In this sense, CNBs, also minimally invasive, could logically be incorporated into this family of standardised reporting systems. Thus, aligning the proposed CNB framework with established models would provide cytopathologists worldwide with a unified reporting system and facilitate a more consistent assessment of malignancy risk across different lesion types, reducing interobserver variability and supporting multicentre research.

One of the core values upheld by the Hippocratic Oath is “to watch over the health and well-being of my patient above all else” [16]. This inherently requires that physicians, regardless of their specialty, remain up to date in knowledge, techniques, technological advancements, and other tools in order to provide the best possible care to their patients [4,5]. It is worth noting that on October 3rd, 2024, the Hospital Universitario del Henares was awarded the EFQM 400+ Seal of Excellence in Management at the European level, reflecting its commitment to continuous improvement and high-quality patient care [17]. This recognition also highlights the institution’s philosophy of placing the patient at the centre of care, a goal to which the interventional pathology consultation actively contributes.

Ultrasound is a well-established diagnostic tool with a long-standing presence in routine clinical practice across multiple medical specialties. Its non-invasive nature, real-time results, and relative technical simplicity have supported its widespread adoption, enabling a rapid and accessible learning curve. Targeted training in the use and interpretation of POCUS has significantly contributed to professional development across various fields of medicine, enhancing clinicians’ diagnostic capabilities and supporting more timely and evidence-based decision-making. This evolution has had a direct impact on delivering more efficient, accurate, and patient-centred care [18]. Today, ultrasound is regarded not only as a diagnostic modality, but also as an extension of modern physical examination, increasingly integrated into clinical settings such as primary care, emergency medicine, intensive care, and rehabilitation. Ultrasonography has represented a major advance in diagnostic imaging and has evolved into highly portable and user-friendly devices. This technological progress has enabled its integration into routine clinical workflows, not only by radiologists or clinicians in acute care settings, but increasingly even by professionals in traditionally non-imaging-focused specialties, such as pathology.

The social perception of the pathologist’s role in medical practice remains limited. Both patients and professionals from other medical specialties report only a partial understanding of the functions and competencies associated with this discipline. This situation contrasts with the historical and scientific significance of pathology, exemplified by Santiago Ramón y Cajal, who, in addition to being widely recognised for his contributions to neuroscience, developed his career as a pathologist and is one of only two Spanish-born recipients of the Nobel Prize in Physiology or Medicine. Beyond its historical importance, pathology is indispensable in contemporary healthcare. The pathologist plays a central role in the diagnostic process, providing essential information not only for establishing the nature of disease but also for guiding prognosis and therapeutic decision-making. Advances in molecular biology, immunohistochemistry, and digital technologies have expanded the scope of pathology, enabling more precise and personalised approaches to patient care. Moreover, pathologists are integral members of multidisciplinary teams, where their expertise informs clinical discussions and contributes to optimising treatment strategies. Pathology is an exceptionally broad specialty that interacts with all other medical disciplines and, therefore, must evolve alongside them to advance our own field [6]. Incorporating ultrasound-guided FNA and CNB into our routine skill set would further elevate our role in diagnosis and align us with the model of the clinical physician, offering an integrated diagnosis and working in close collaboration with colleagues from other specialties [3,10]. This would position the pathologist as a multifaceted and multidisciplinary professional [3].

A common concern when adopting this practice is the potential for professional overlap with radiologists. However, it is important to clarify that in Spain, there is no legal concept of “professional intrusion” between medical specialties [6]. In fact, dermatologists and endocrinologists already use ultrasound in their clinics and may even examine their biopsies and smear preparations under the microscope [4]. Nevertheless, it must be emphasised that the goal of the pathologist in using ultrasound is not to make an imaging diagnosis, but rather to integrate macroscopic features into the diagnostic process and accurately guide the needle to the target area for sampling [3]. The only legal requirement to perform a procedure is that the physician must be adequately trained and sufficiently skilled to do so competently [5]. To this end, training programmes are available: the College of American Pathologists, recognising the growing interest in this area, offers courses in ultrasound-guided FNA and CNB [2]. There are also rotation programmes and fellowships for both pathology residents and consultants in specialised centres [8,9,10]. Training is typically initiated using phantom simulation models to acquire technical skills before progressing to supervised procedures in the clinic [6].

Another issue that may concern the pathologists is how to combine microscopic diagnostic work with interventional consultation, or if there is a need for a large staff to carry out both functions. Nevertheless, based on our experience, with proper department organization, this scenario is feasible. Additionally, as more pathologists acquire interventional competencies and more clinicians recognize the utility of the consultation, the number of specialized personnel in this area will increase, allowing for the establishment of rotating shifts within the department, always adapting to the specific situation and needs of each service.

There are several advantages to having pathologists perform these procedures. Firstly, it improves patient comfort and perceived quality of care, by reducing the number of hospital visits and enabling faster diagnosis [4,6,9]. This new clinical pathway also reduces costs for both the hospital and the healthcare system, while optimising appointment schedules and decreasing the demand for other diagnostic tests [5,6]. Furthermore, it enables more precise and personalised medicine—an increasingly central objective in modern medical practice [4]. This pursuit of personalised care is further enhanced by the ability to perform molecular and immunohistochemical studies on samples obtained during interventional pathology consultations.

Over time, medicine has progressively advanced towards diagnostic precision and personalised care across all its domains. One of the most widely adopted, in-demand, and innovative developments has been the incorporation of molecular studies capable of identifying genetic alterations involved in neoplastic processes, which may also serve as therapeutic targets for patients. Molecular biology has been driving the transformation of medicine for several years now, and with it, the evolution of anatomical pathology. The term “molecular cytopathology” was coined back in 2004, and over the past decade, it has been progressively refined [19]. Today, the combination of cytomorphology, immunocytochemistry, and molecular testing in cytology plays a leading role in clinical decision-making regarding patient management [19]. This shift towards a morphomolecular model within our discipline is now well established, particularly in tumours such as lung cancer and in thyroid, this last one falls within the scope of the interventional pathologist.

Beyond the routine analysis of biomarkers in cytological samples, increasingly ambitious and realistic goals are emerging, such as the use of next-generation sequencing (NGS). Its incorporation into routine practice using tissue samples is already a reality, and its application to cytological specimens is becoming ever more feasible. In fact, the molecular cytopathology community is actively working on the development of international networks aimed at validating NGS and promoting its global applicability. The objective is to demonstrate the reliability of cytological samples for any type of genomic analysis [19]. However, the applicability of such studies is often contingent upon the presence of a minimum threshold of cellularity that is representative of the lesion. Frequently, these analyses must be postponed until surgical specimens become available, thereby delaying therapeutic decision-making. Nevertheless, the feasibility of performing these studies on samples obtained via CNB in interventional pathology clinics could significantly expedite the process. This is made possible through the use of the Touch Print ROSE technique, which ensures sufficient cellular representation within the sample.

Combined with the implementation of rapid-access nodule assessment clinics within hospitals, this approach would enable pathologists to convey diagnostic findings to the treating physician more swiftly, thus facilitating earlier initiation of the most appropriate treatment for the patient. A clear example of the value of performing ancillary studies on samples obtained during interventional pathology consultations, particularly given that our main field of expertise is the head and neck region, is the immunohistochemical evaluation of PD-L1 expression. This marker directly influences the therapeutic approach for patients eligible for monoclonal antibody treatments. This point further reinforces the integral role of the interventional pathologist, who once again stands as the only professional capable of integrating the macroscopic ultrasound image, the on-site validation of the obtained material for additional studies, the cytomorphology, and the results of immunocytochemistry and molecular testing.

In this context, the adoption of standardized classification systems has proven to enhance diagnostic communication between pathologists and clinicians, and to facilitate the comparison of results across institutions. The *Sydney System* for lymph node cytopathology, proposed by Al-Abbadi et al. [20], provides an internationally recognized framework that defines clear diagnostic categories associated with a risk of malignancy, supporting more consistent clinical management. Its applicability and diagnostic reliability in ultrasound-guided procedures performed by interventional pathologists have subsequently been validated in independent series [21]. In line with these principles, our proposed histological classification for lymph node core-needle biopsy (CNB) aims to establish a reproducible and standardized approach that strengthens cyto-histologic correlation and optimizes clinical decision-making.

## 5. Conclusions

Pathologists with appropriate training can successfully perform ultrasound-guided FNA and CNB procedures of superficial lymph node and non-nodal lesions, including immediate ROSE in the clinic, achieving excellent diagnostic correlation. This approach reduces the number of passes and smear slides that burden the laboratory, decreases healthcare costs, enhances patients’ perceived quality of care, enables personalized medicine, and establishes the pathologist as a key figure in minimally invasive diagnosis.

Given the strong diagnostic correlation observed, unifying the classification system for lymph node lesions in CNB with the Sydney system would improve communication among pathologists and with other medical specialists.

## Figures and Tables

**Figure 1 diagnostics-15-02628-f001:**
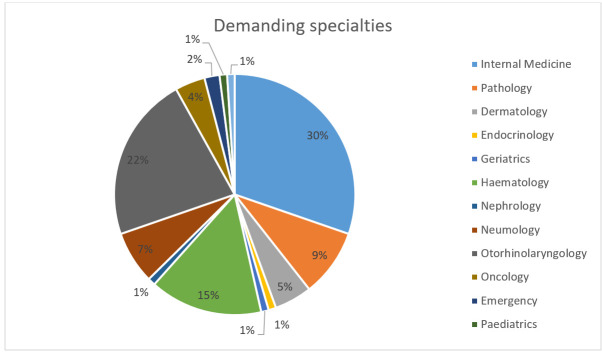
Medical specialties that referred patients to the interventional pathologist consultation.

**Figure 2 diagnostics-15-02628-f002:**
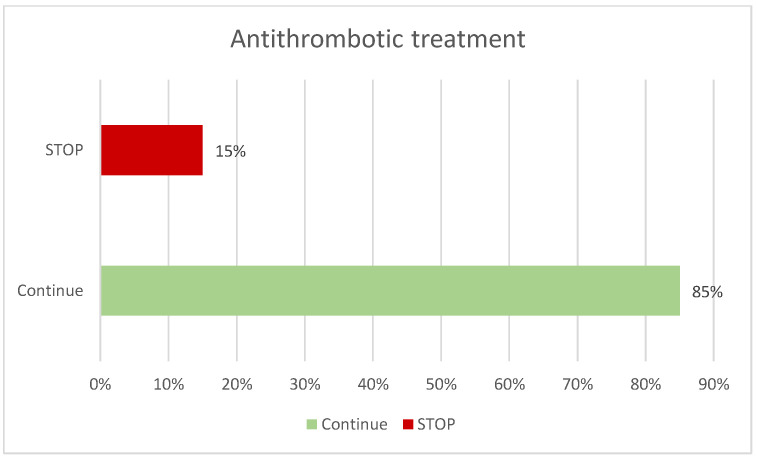
Management of antithrombotic treatment.

**Figure 3 diagnostics-15-02628-f003:**
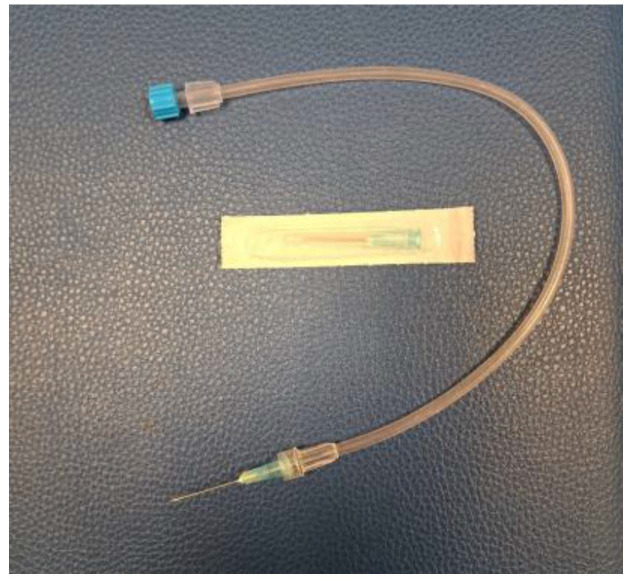
Microlance 23G needle.

**Figure 4 diagnostics-15-02628-f004:**
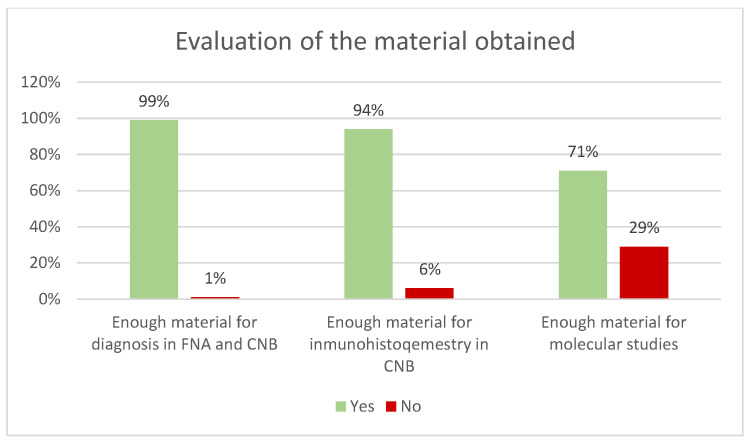
Evaluation of the material obtained for diagnosis and complementary techniques.

**Figure 5 diagnostics-15-02628-f005:**
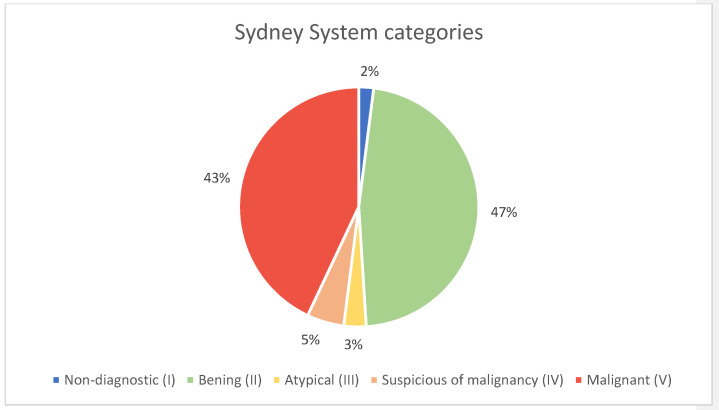
Diagnosis based in the categories of the 2020 Sydney System.

**Figure 6 diagnostics-15-02628-f006:**
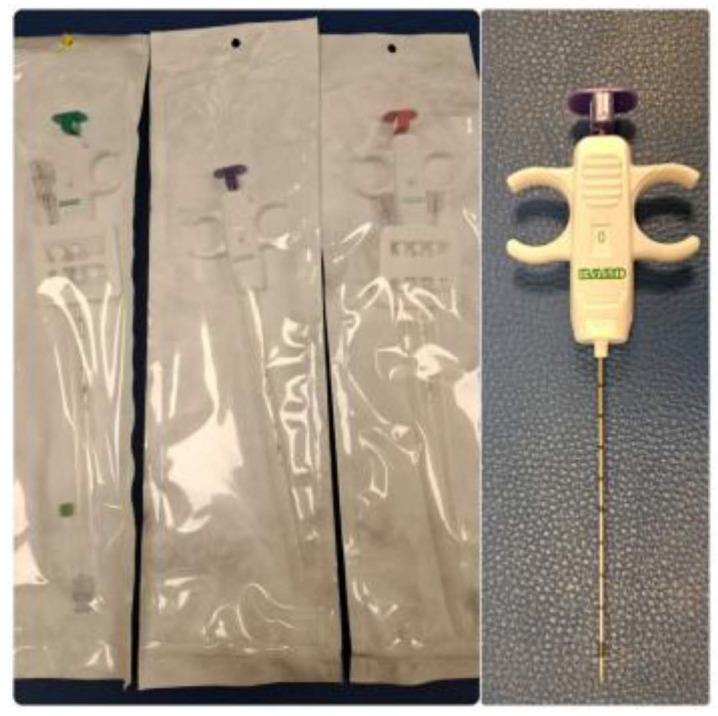
CNB dispositives. 14G, 16G and 18G calibers.

**Figure 7 diagnostics-15-02628-f007:**
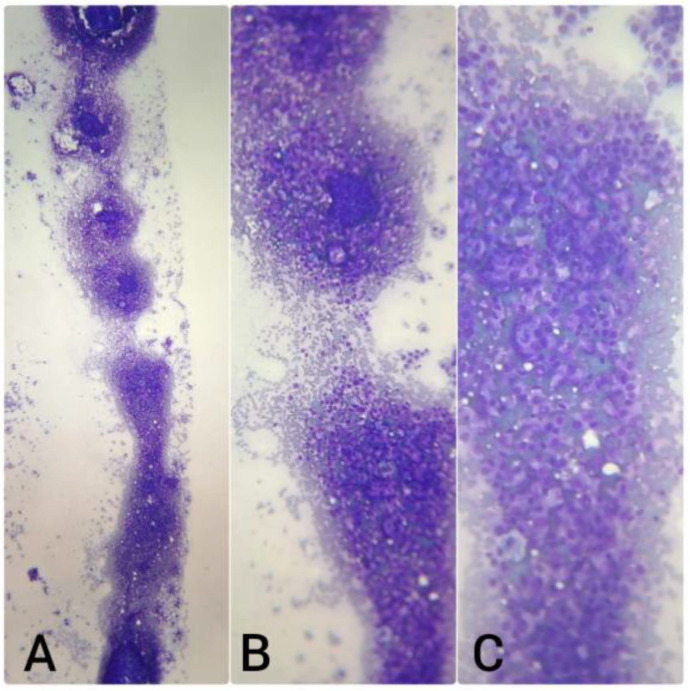
Touch print ROSE of a CNB in a metastasic lymph node. (**A**) 40X, (**B**) 100X, (**C**) 200X.

**Figure 8 diagnostics-15-02628-f008:**
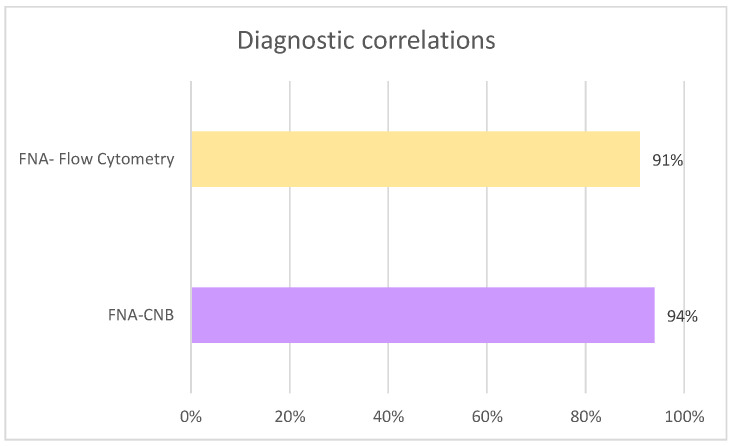
Concordance in diagnosis.

**Table 1 diagnostics-15-02628-t001:** Descriptive distribution of diagnostic categories and complementary procedures.

Variable	Category	Frequency (*n*)	Percent (%)
Sydney System (FNA)	I	2	1.5
	II	63	47
	III	4	3
	IV	7	5.2
	V	58	43.3
CNB	Yes	31	23.1
	No	103	76.9
Flow Cytometry	Yes	79	59
	No	35	26.1
	No necessary	20	14.9
FNA positive (FNA category 4–5)	Yes	65	48.5
	No	69	51.5

**Table 2 diagnostics-15-02628-t002:** Average of passes, slides and cylinders.

	FNA	CNB
Total	134	31
Passes average	1	3
Slides/Cylinders averages	3	4

**Table 3 diagnostics-15-02628-t003:** Contingency table and diagnostic performance of FNA vs. CNB.

	CNB Positive	CNB Negative	
FNA positive	28	0	PPV: 100%
FNA negative	2	1	NPV: 33.3%
	Sensitivity: 93.3%	Specificity: 100%	Total: 31

**Table 4 diagnostics-15-02628-t004:** Contingency table and diagnostic performance of FNA vs. Flow Cytometry.

	**Flow Cytometry Positive**	**Flow Cytometry Negative**	
FNA positive	26	5	PPV: 83.9%
FNA negative	2	46	NPV: 95.8%
	Sensitivity: 92.9%	Specificity: 90.2%	Total: 79

## Data Availability

The data supporting the findings of this study are not publicly available due to ethical and privacy restrictions. Specifically, the dataset contains sensitive patient information and is protected under the approval conditions of the institutional ethics committee. Therefore, access to the data is restricted in order to preserve patient confidentiality and comply with regulatory and ethical guidelines.

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
