# Peer review of "The Importance of the Interventional Pathologist in Fine Needle Aspiration and Core Needle Biopsy Procedures for Lymph Node Lesions: A Retrospective Analysis of Diagnostic Correlation and Proposal of a Classification System for Lymph Node Core Needle Biopsies"

_diagnostics, 2025, doi:10.3390/diagnostics15202628_

Round 1

Reviewer 1 Report

Comments and Suggestions for Authors

Although the manuscript makes an important contribution to the ongoing discussion regarding the evolving role of the clinical and interventional pathologist, and the authors demonstrate technical expertise and up-to-date knowledge, the text presents several critical issues.

First, it is unclear whether the authors intend to present an editorial, a commentary, or an original research article.

Both the title and the stated aim of the paper suggest a proposed classification for CNB. While this is an interesting initiative, the proposal remains underdeveloped, lacking concrete examples, defined categories, and clear criteria. Furthermore, the suggestion to apply the Sydney classification for lymph nodes is insufficient and not adequately justified.

The statistical analysis is weak or entirely absent, with the exception of the calculation of positive predictive value. The significance of the results is not discussed in depth. The only outcomes reported are sample adequacy and the concordance between FNA and CNB. While these are important parameters, their interpretation is unclear, and the overall message of the study remains ambiguous.

The anatomical target is also poorly defined. Referring only to “nodular lesions” is overly vague and limiting. The specific diagnoses obtained from the samples are not clearly reported. Additionally, the limitations of each method—FNA and CNB—are not addressed.

Comments on the Quality of English Language

Overall, the manuscript is lengthy and at times redundant, with long and complex sentences that compromise readability.

Author Response

Dear Revisor 1,

I sincerely appreciate your valuable comments, which have undoubtedly contributed to enhancing the quality of the manuscript and making it more suitable for publication in the journal. I have carefully addressed all of your suggestions and have incorporated the corresponding changes into the revised manuscript, where they are highlighted in bold for clarity. For your convenience, I have also provided a point-by-point response below, following the same order as your comments.

  • First, it is unclear whether the authors intend to present an editorial, a commentary, or an original research article.

- The specification that this is an original research article has been added to the first sentence of the first paragraph of the Materials and Methods section.

  • Both the title and the stated aim of the paper suggest a proposed classification for CNB. While this is an interesting initiative, the proposal remains underdeveloped, lacking concrete examples, defined categories, and clear criteria. Furthermore, the suggestion to apply the Sydney classification for lymph nodes is insufficient and not adequately justified.

- A paragraph has been added to the Materials and Methods section clarifying which Sydney system was used. The tittle and the abstract have been also modified.

  • The statistical analysis is weak or entirely absent, with the exception of the calculation of positive predictive value. The significance of the results is not discussed in depth. The only outcomes reported are sample adequacy and the concordance between FNA and CNB. While these are important parameters, their interpretation is unclear, and the overall message of the study remains ambiguous.

- The statistical analysis has been improved and expanded in both the Materials and Methods section and the Discussion

  • The anatomical target is also poorly defined. Referring only to “nodular lesions” is overly vague and limiting. The specific diagnoses obtained from the samples are not clearly reported. Additionally, the limitations of each method—FNA and CNB—are not addressed.

- The database has been re-evaluated and errors corrected. The sample corresponds entirely to lymph node lesions, as detailed in the Materials and Methods section. Nevertheless, although these cases are not part of the study sample, our centre also has experience performing FNA of superficial extranodal lesions, which is reflected in an additional paragraph included in the Discussion. Similarly, this point has been clarified in the title of the article as well as in the abstract.

- A paragraph has been added clarify the specific diagnoses obtained from the samples that had not been clearly reported.

            -The limitations of the FNA are reported in the introduction.

-The limitation regarding   the number of CNBs has been addressed in the discussion section

  • Overall, the manuscript is lengthy and at times redundant, with long and complex sentences that compromise readability.

- Both the English language and the writing style have been revised, with the aim of making the sentences clearer and improving the overall readability of the manuscript.

We hope these changes meet your expectations. In addition to the proposed modifications, the title and abstract have been revised, the results section has been reviewed and updated, new graphs have been created (Table 1,3,4), modifying others (figure 3,4,6 and table 2), and two bibliographic references have been added. Thank you very much for your time and consideration.

Reviewer 2 Report

Comments and Suggestions for Authors

This is an interesting manuscript but could be substantially improved by the following:

1) From reading this manuscript it is not clear what types of nodules the authors are addressing. Were these nodules arising in lymph nodes or were they from all sites including thyroid, pancreas, liver, lung, etc.? This should be clearly stated in their Introduction and Materials & Methods.

2) The authors use the Sydney System for Classification. There are several Sydney systems and they have not specified which one they are using. There is a Sydney System for Gastrointestinal Biopsies and a Sydney System for Reporting Lymphoproliferative Disease. The Sydney System is not to this reviewer’s knowledge used for other types of lesions such as thyroid, metastatic carcinoma, etc. The authors need to clarify which Sydney System they are using and how they are utilizing it. 

3) The authors discuss the use of fine-needle aspiration and core-needle biopsy in their review. Currently, fine-needle biopsy is being increasingly utilized. FNB uses a 22 to 25-gauge needle, in most cases identical in gauge to FNA needles. Thus, FNB should be discussed as it is being increasingly utilized and, in some cases, replacing traditional core needle biopsy (14-18 gauge). Therefore, the authors should address this technique for tissue acquisition. 

4) In their practice the authors appear to send pathologists for on-site FNA performance and rapid on-site evaluation. This indicates that they have sufficient cytopathologists and/or cytotechnologists to supply on-site staffing. In many if not most pathology/cytopathology practices, adequate numbers of cytopathologists/cytotechnologists are not available to staff on-site clinics or other sites for the performance of ROSE or even the performance of ultrasound guided FNAs. The authors need to address this issue.

5) While the study has a large experience in FNA material, there are relatively few core needle biopsies limiting the ability to adequately compare FNA and core biopsy. The authors should address this issue.

6) In many institutions, core biopsies are reviewed by the surgical pathology team with or without input from cytopathologists. Surgical pathologists review core biopsy specimens in a method similar to that of other surgical pathology specimens. Whenever possible, they give a definitive diagnosis of the lesion present (adenocarcinoma, granulomatous disease, reactive changes). They do not use a grading system and probably would be uncomfortable adopting such a system. The authors should more fully discuss what a 5-tier grading system would do to improve surgical pathology diagnoses of core needle biopsies. 

7) The order of the manuscript sections does not follow standard sequence. In most cases, the sequence would be: 1. Introduction 2. Materials & Methods 3. Results and 4. Discussion and Conclusions. That format is not followed by the authors. They should comply with standard practice. 

Author Response

Dear Revisor 2,

I sincerely appreciate your valuable comments, which have undoubtedly contributed to enhancing the quality of the manuscript and making it more suitable for publication in the journal. I have carefully addressed all of your suggestions and have incorporated the corresponding changes into the revised manuscript, where they are highlighted in bold for clarity. For your convenience, I have also provided a point-by-point response below, following the same order as your comments.

  • From reading this manuscript it is not clear what types of nodules the authors are addressing. Were these nodules arising in lymph nodes or were they from all sites including thyroid, pancreas, liver, lung, etc.? This should be clearly stated in their Introduction and Materials & Methods.

- The database has been re-evaluated and errors corrected, with the sample now confirmed to consist entirely of lymph node lesions, as detailed in the Materials and Methods section. Although not part of this study, our centre also has experience performing FNA of superficial extranodal lesions, which is reflected in an additional paragraph in the Discussion. To improve clarity, the Introduction now includes a paragraph specifying the types of lesions accessible for FNA and the clinical settings in which these procedures can be performed, while the Materials and Methods section, title, and abstract have been updated to indicate the types and anatomical locations of lesions included in this study..

  • The authors use the Sydney System for Classification. There are several Sydney systems and they have not specified which one they are using. There is a Sydney System for Gastrointestinal Biopsies and a Sydney System for Reporting Lymphoproliferative Disease. The Sydney System is not to this reviewer’s knowledge used for other types of lesions such as thyroid, metastatic carcinoma, etc. The authors need to clarify which Sydney System they are using and how they are utilizing it. 
  • A paragraph has been added in the Materials and Methods section to clarify this point as well as in the abstract.

  • The authors discuss the use of fine-needle aspiration and core-needle biopsy in their review. Currently, fine-needle biopsy is being increasingly utilized. FNB uses a 22 to 25-gauge needle, in most cases identical in gauge to FNA needles. Thus, FNB should be discussed as it is being increasingly utilized and, in some cases, replacing traditional core needle biopsy (14-18 gauge). Therefore, the authors should address this technique for tissue acquisition. 

- Thank you for your valuable suggestion regarding fine-needle biopsy (FNB). It is indeed an interesting and promising technique. However, unfortunately, in our setting, and therefore in this study, we do not have access to this method and currently perform only CNB using calibres 14 to 18. Nevertheless, we will certainly consider and study the implementation of FNB for future research. A statement to that effect has been included in the Materials and Methods section.

  • In their practice the authors appear to send pathologists for on-site FNA performance and rapid on-site evaluation. This indicates that they have sufficient cytopathologists and/or cytotechnologists to supply on-site staffing. In many if not most pathology/cytopathology practices, adequate numbers of cytopathologists/cytotechnologists are not available to staff on-site clinics or other sites for the performance of ROSE or even the performance of ultrasound guided FNAs. The authors need to address this issue.

- Regarding this point, indeed, both the pathologist and the technician are responsible for performing the FNA and therefore must temporarily leave their microscopic diagnostic duties to attend the consultation. In the case of the Hospital Universitario del Henares, where this study was conducted, the consultation room was located within the same department, just a few meters from the pathologist’s office. This proximity greatly facilitates time management and compatibility with microscopy work. At this hospital, there is only one pathologist responsible for performing both FNAs and CNBs, and a rotating system of technicians with appropriate training assists the physician. As you can see in our sample, the amount of patients attended in our centre were approximately 65 per year, a volume manageable for a second-level hospital.

Thus, as more pathologists acquire interventional competencies and more clinicians become aware of the consultation’s usefulness, the number of specialized personnel in this area within anatomical pathology departments across hospitals of all levels will increase. It could even be considered to rotate consultation days among interventional pathologists in the service, dedicating one morning to patients and reorganizing diagnostic work among the other team members. The same could be done with technicians. Of course, this would depend on the specific situation and organizational structure of each centre; however, we encourage considering this possibility to provide patients with a high-quality service.

This aspect has been clarified in a paragraph in the Materials and Methods section and Discussin.

  • While the study has a large experience in FNA material, there are relatively few core needle biopsies limiting the ability to adequately compare FNA and core biopsy. The authors should address this issue.

- This point has been addressed and presented as a limitation in the discussion section

  • In many institutions, core biopsies are reviewed by the surgical pathology team with or without input from cytopathologists. Surgical pathologists review core biopsy specimens in a method similar to that of other surgical pathology specimens. Whenever possible, they give a definitive diagnosis of the lesion present (adenocarcinoma, granulomatous disease, reactive changes). They do not use a grading system and probably would be uncomfortable adopting such a system. The authors should more fully discuss what a 5-tier grading system would do to improve surgical pathology diagnoses of core needle biopsies. 

- A paragraph regarding this has been added in the discussion section

- In the penultimate paragraph of the discussion, we address the use of NGS on material obtained through CNB instead of surgical biopsy. This approach allows us to obtain molecular results in a minimally invasive manner, with reduced risk and discomfort for the patient.

  • The order of the manuscript sections does not follow standard sequence. In most cases, the sequence would be: 1. Introduction 2. Materials & Methods 3. Results and 4. Discussion and Conclusions. That format is not followed by the authors. They should comply with standard practice. 

- The order has been corrected; it appears to have been an error that occurred during the manuscript submission process.

We hope these changes meet your expectations. In addition to the proposed modifications, the title and abstract have been revised, the results section has been reviewed and updated, new graphs have been created (Table 1,3,4), modifying others (figure 3,4,6 and table 2), and two bibliographic references have been added. Thank you very much for your time and consideration.

Additionally, the statistical analysis has been improved and expanded in both the Materials and Methods section and the Discussion

Round 2

Reviewer 1 Report

Comments and Suggestions for Authors

The changes made have significantly improved the manuscript.